# Gene activation by a CRISPR-assisted *trans* enhancer

**Xinhui Xu, Jinliang Gao, Wei Dai, Danyang Wang, Jian Wu, Jinke Wang\***

State Key Laboratory of Bioelectronics, Southeast University, Nanjing, China

**Abstract** The deactivated CRISPR/Cas9 (dCas9) is now the most widely used gene activator. However, current dCas9-based gene activators are still limited by their unsatisfactory activity. In this study, we developed a new strategy, the CRISPR-assisted *trans* enhancer, for activating gene expression at high efficiency by combining dCas9-VP64/sgRNA with the widely used strong CMV enhancer. In this strategy, CMV enhancer DNA was recruited to target genes in *trans* by two systems: dCas9-VP64/csgRNA-sCMV and dCas9-VP64-GAL4/sgRNA-UAS-CMV. The former recruited *trans* enhancer by annealing between two short complementary oligonucleotides at the ends of the sgRNA and *trans* enhancer. The latter recruited *trans* enhancer by binding between GAL4 fused to dCas9 and UAS sequence of *trans* enhancer. The *trans* enhancer activated gene transcription as the natural looped *cis* enhancer. The *trans* enhancer could activate both exogenous reporter genes and variant endogenous genes in various cells, with much higher activation efficiency than that of current dCas9 activators.

DOI: https://doi.org/10.7554/eLife.45973.001

## Introduction

Clustered regularly interspaced short palindromic repeats (CRISPR) was originally identified in the immune system of bacteria, with the function of destroying the invading microphage DNA by enzymatic digestion. The system has been developed into a highly efficient gene editing tool (*Doudna and Charpentier, 2014*; *Mali et al., 2013c*), and also into new gene activators. For example, the dead Cas9 (dCas9) and its associated single guide RNA (sgRNA) have been widely used to regulate gene expression in recent years (*Dominguez et al., 2016*; *Hilton et al., 2015*; *Jinek et al., 2012*; *Kiani et al., 2015*; *Mali et al., 2013b*; *Radzisheuskaya et al., 2016*; *Wang et al., 2016*). Both dCas9 and sgRNA have been engineered for activating or repressing gene expression. For instance, the dCas9 protein has been fused with various gene activation or repression domains, such as VP48 (*Cheng et al., 2013*), VP160 (*Perrin et al., 2017*), VP64 (*Maeder et al., 2013*; *Perez-Pinera et al., 2013*), VPR (VP64-p65-Rta) (*Chavez et al., 2015*), and KRAB (*Zheng et al., 2018*). Additionally, the dCas9 protein has been fused with other functional domains with transcriptional regulatory functions, such as p300 (*Hilton et al., 2015*), LSD1 (*Kearns et al., 2015*), Dnmt3a (*Liu et al., 2016a*; *Saunderson et al., 2017*), and Tet1 (*Choudhury et al., 2016*; *Liu et al., 2016a*). Based on these domains, more elaborate activators have been developed for more potent activation of target genes in mammalian cells, such as SunTag (dcas9-GCN4/sgRNA plus scFV-VP64) (*Tanenbaum et al., 2014*) and SPH (dCas9-GCN4/sgRNA plus scFV-p65-HSF1) (*Zhou et al., 2018*). Furthermore, some inducible dCas9 systems have been developed to control activity of dCas9 activators in cells, such as light-activated CRISPR/Cas9 effector (*Nihongaki et al., 2015*; *Polstein and Gersbach, 2015*), hybrid drug inducible CRISPR/Cas9 technology (HIT) (*Lu et al., 2018*), and CRISPR activator gated by human antibody-based chemically induced dimerizers (AbCIDs) (*Liu et al., 2018*). However, it is difficult to package most of these dCas9 fusion proteins into adeno-associated virus (AAV) for in vivo application.

**\*For correspondence:**
wangjinke@seu.edu.cn

**Competing interests:** The authors declare that no competing interests exist.

sgRNA has also been engineered to develop new dCas9-based activators. Compared with dCas9 engineering, sgRNA is more simple, flexible, and efficient to redesign. Moreover, the engineered sgRNA is more helpful for the in vivo application of dCas9-based activators because of its limited length for virus packaging. The most widely used sgRNA-based gene activator is the synergistic activation mediator (SAM) system, in which MS2 loops are fused to the 3′ end of sgRNA (*Konermann et al., 2015*; *Liao et al., 2017*). Similarly, Pumilio/FBF (PUF), modular scaffold RNAs (MS2, PP7, and com), and riboswitches have been fused to sgRNA (*Cheng et al., 2016*; *Liu et al., 2016b*; *Zalatan et al., 2015*). However, these chimeric sgRNA-based strategies were limited by their complicated RNA aptamers and the cognate RNA-binding fusion proteins.

Although variant dCas9-based activators have been developed (*Chen and Qi, 2017*), the current dCas9-based transcriptional activators are relatively inefficient in endogenous gene activation and cell reprogramming (*Gao et al., 2014*). By a systematic comparison of relative potency and effectiveness across various cell types and species (human, mouse, and fly) (*Chavez et al., 2016*), it was found that the majority of second-generation activators had higher activity than that of dCas9-VP64, with the three most potent activators being VPR, SAM, and Suntag. The three activators were consistently better than VP64 across a range of target genes and cellular environments. Moreover, the three activators showed a similar level of activity, and fusing their elements did not yield more potent activators (*Chavez et al., 2016*). Novel, more potent dCas9-based activators might be built by creating other architectures.

Almost three decades ago, the human cytomegalovirus (CMV) enhancer/promoter (referred to as CMV enhancer hereafter) was found. It is a natural mammalian promoter with high transcriptional activity (*Boshart et al., 1985*). Later studies showed the CMV enhancer to be a strong enhancer in various mammalian cells (*Boshart et al., 1985*; *Foecking and Hofstetter, 1986*; *Ho et al., 2015*; *Kim et al., 1990*). This enhancer has been widely used to drive ectopic expression of various genes in a wide range of mammalian cells, and to drive ectopic expression of exogenous genes in broad tissues in transgenic animals (*Furth et al., 1991*; *Schmidt et al., 1990*), protein production by gene engineering, and gene therapy. We have recently improved the transcriptional activity of the CMV enhancer by changing the natural NF-κB binding sites into artificially selected NF-κB binding sequences with high binding affinity (*Wang et al., 2018*). Therefore, we conceived that a unique architecture may be constructed to improve dCas9-based activators using the CMV enhancer.

In this study, mimicking the natural enhancer activating gene expression by a loop structure (*Carter et al., 2002*; *Tolhuis et al., 2002*), we developed a new dCas9-based activator by combining dCas9/sgRNA with CMV enhancer. The 3′ end of sgRNA was redesigned to contain a short capture sequence complementary to a stick-end of a double-stranded CMV enhancer. The CMV enhancer was anchored to the promoter region of a target gene by dCas9/sgRNA. The dCas9/sgRNA-recruited CMV enhancer thus functioned like a natural looped *cis* enhancer in a *trans* form. We found that the new activator could efficiently activate exogenous and endogenous genes in various cells. More importantly, the CMV enhancer could be also recruited to a target gene in *trans* using another system consisting of dCas9-VP64-GAL4/sgRNA and UAS-CMV.

## Results

### Principle of gene activation by a CRISPR-assisted *trans* enhancer

The principle of activating gene expression by a CRISPR-assisted *trans* enhancer is schematically illustrated in *Figure 1a*. A capture sgRNA (csgRNA) was produced by adding a capture sequence to the 3′ end of a normal sgRNA sequence. A linear stick-end CMV (sCMV) enhancer was produced by adding a 3′ end single-strand overhang. The overhang can anneal with the csgRNA capture sequence. When dCas9 protein was guided to the promoter of the target gene by csgRNA, sCMV could be recruited by csgRNA. The recruited sCMV may activate the transcription of the target gene like a natural looped *cis* enhancer. Because the dCas9/csgRNA-anchored sCMV functions as a transcription factors in *trans*, we named it a *trans* enhancer to distinguish it from the natural *cis* enhancer.

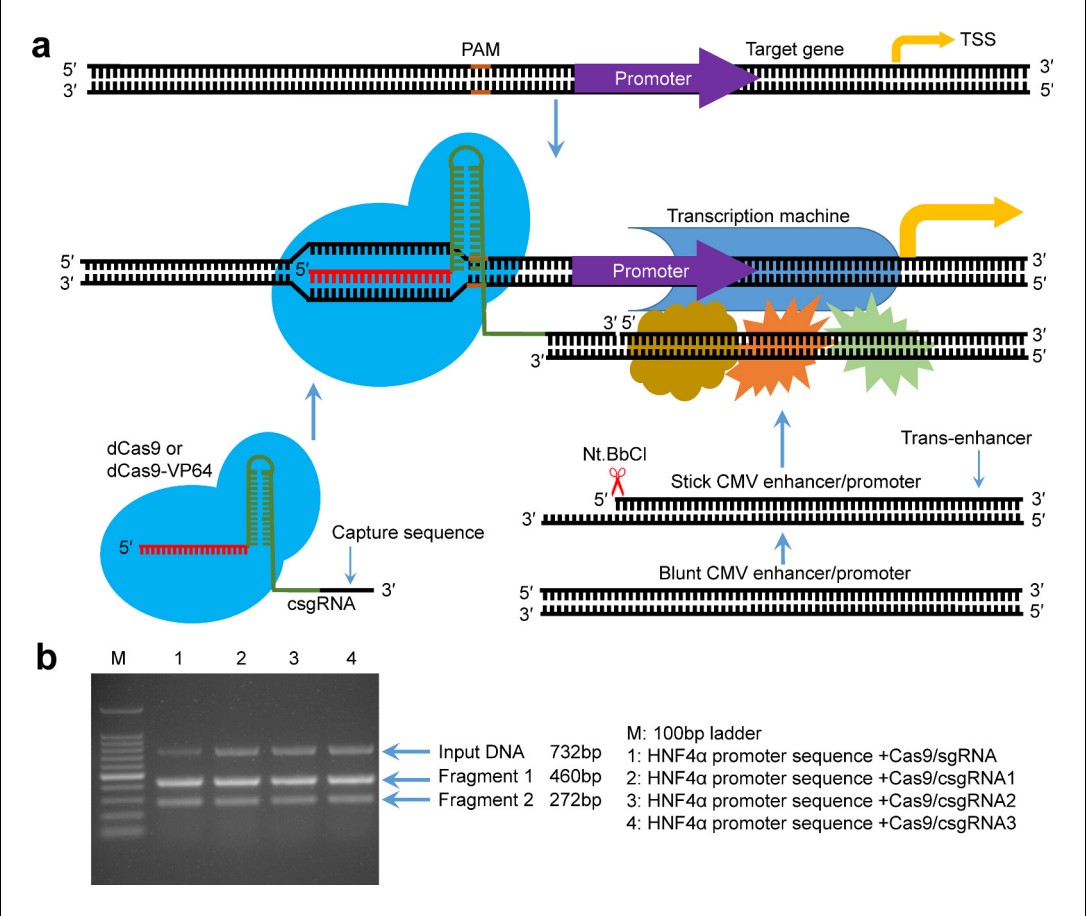

**Figure 1.** Principle of gene expression activation by the CRISPR-assisted *trans* enhancer and evaluation of designed csgRNAs. (**a**) Schematic illustration of the principle of gene expression activation by the CRISPR-assisted *trans* enhancer. A capture sequence is added to the 3′ end of sgRNA, which is used to capture a *trans* CMV enhancer with a single-stranded overhang that can anneal with the capture sequence of sgRNA. The captured *trans* CMV enhancer may function like the natural looped *cis* enhancer to activate transcription of the gene of interest, including exogenous and endogenous genes. (**b**) In vitro target DNA cutting by the Cas9-csgRNA complex. DNA fragments (732 bp) amplified from the HNF4α promoter region were, respectively, cut by the Cas9/csgRNA and Cas9/sgRNA complexes. csgRNA1, csgRNA2 and csgRNA3 had the same target sequence but different capture sequences.

DOI: https://doi.org/10.7554/eLife.45973.002

## Effect of capture sequence on the function of sgRNA

To determine whether the capture sequence affects the function of sgRNA, we prepared a normal sgRNA and three csgRNAs targeting the same site of the HNF4α promoter. The three csgRNAs had different capture sequences. We used these sgRNAs to associate with the Cas9 endonuclease to cut a 732 bp HNF4α promoter fragment. The results indicated that the target DNA could be digested by all sgRNAs (*Figure 1b*), indicating that the capture sequence did not affect the sgRNA function.

## Activation of exogenous reporter gene by *trans* enhancer

To determine whether the CRISPR-assisted *trans* enhancer activates gene expression, we constructed a reporter construct of HNF4α promoter (pEZX-HP-ZsGreen). 293 T cells were then transfected with various vectors (*Figure 2a*, *Figure 2—figure supplement 1*). The transfection indicated that ZsGreen expression could be successfully activated by dCas9/csgRNA2-sCMV but not activated by dCas9/csgRNA2-blunt CMV (bCMV). Although the dCas9/csgRNA2-sCMV showed a similar activation level to Cas9-VP64/sgRNA, it was far inferior to *cis* CMV enhancer. To improve the

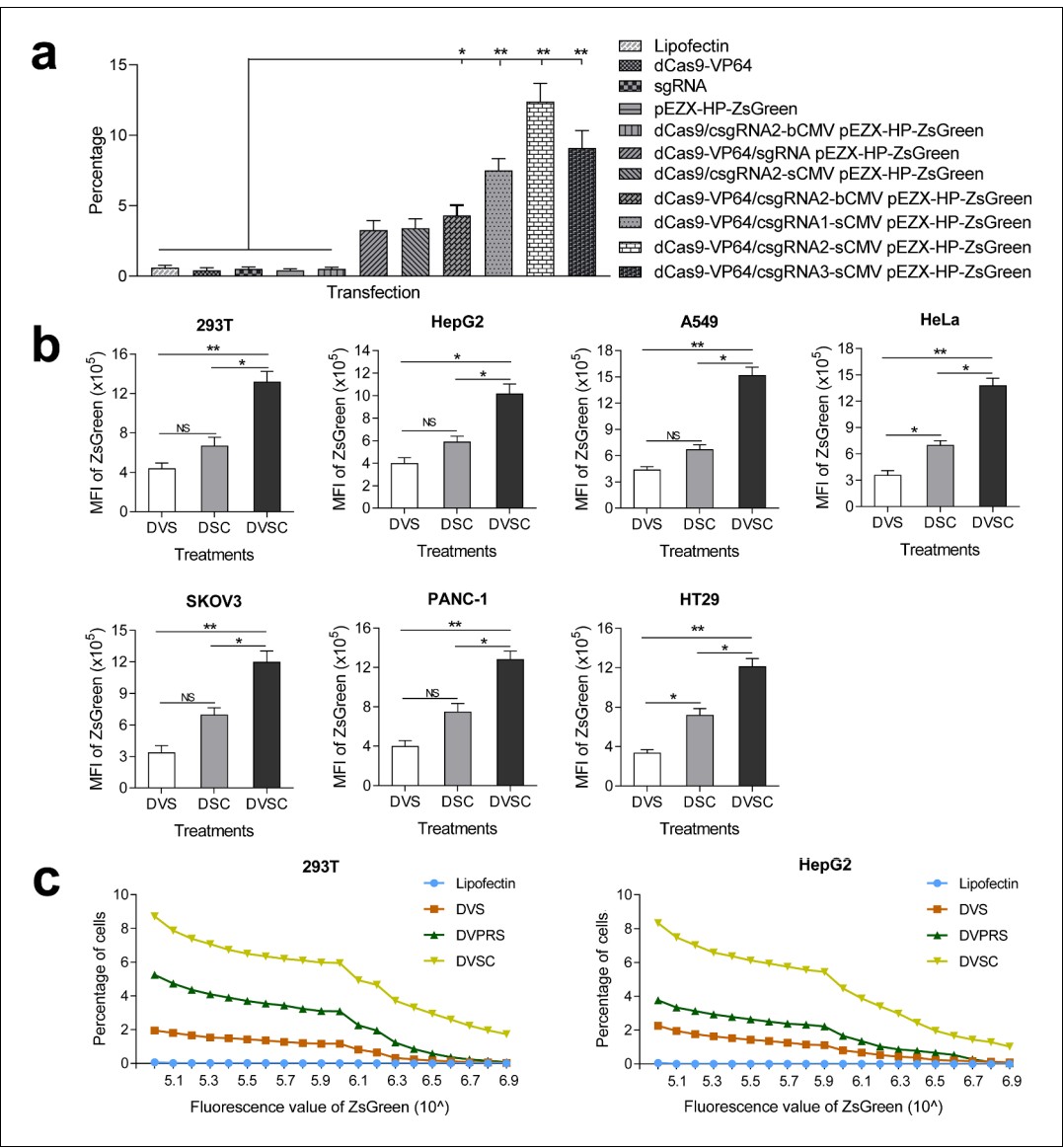

**Figure 2.** Activation of an exogenous reporter gene ZsGreen under the control of a HNF4α promoter by the CRISPR-assisted *trans* enhancer in multiple cells. (a) Transcriptional activation of reporter gene ZsGreen in various cells transfected by different vectors. The florescence intensity of cells was analyzed by flow cytometry and is shown as the mean fluorescence intensity (MFI). Transfections: DVS, dCas9-VP64/sgRNA; DSC, dCas9/csgRNA-sCMV; DVSC, dCas9-VP64/csgRNA-sCMV. (b) Comparison between *trans* enhancer and VPR. Cells were transfected with three different transcriptional activation systems to activate reporter gene ZsGreen. The florescence intensity of cells was analyzed by flow cytometry and the number of cells with certain fluorescence intensity was counted. Transfections: Lipo, lipofectin; DVS, dCas9-VP64/sgRNA; DVPRS; dCas9-VPR/csgRNA; DVSC, dCas9-VP64/csgRNA-sCMV.

DOI: https://doi.org/10.7554/eLife.45973.003

The following figure supplements are available for figure 2:

**Figure supplement 1.** Activation of an exogenous reporter gene ZsGreen under the control of a HNF4α promoter by the CRISPR-assisted *trans* enhancer in 293 T cells.

DOI: https://doi.org/10.7554/eLife.45973.004

**Figure supplement 2.** Activation of an exogenous reporter gene ZsGreen under the control of a HNF4α promoter by the CRISPR-assisted *trans* enhancer in HepG2 and PANC1 cells.

DOI: https://doi.org/10.7554/eLife.45973.005

**Figure supplement 3.** Activation of an exogenous reporter gene ZsGreen under the control of a HNF4α promoter by the CRISPR-assisted *trans* enhancer in A549 and HeLa cells.

*Figure 2 continued on next page*

*Figure 2 continued*

DOI: https://doi.org/10.7554/eLife.45973.006

**Figure supplement 4.** Activation of an exogenous reporter gene ZsGreen under the control of a HNF4α promoter by the CRISPR-assisted *trans* enhancer in SKOV3 and HT29 cells.

DOI: https://doi.org/10.7554/eLife.45973.007

performance of *trans* CMV, we tried transfecting 293 T cells with dCas9-VP64/csgRNA2-sCMV. The results indicated that ZsGreen expression was highly activated by the transfection. In contrast, the dCas9-VP64/csgRNA2-bCMV showed a similar activation level to dCas9-VP64/sgRNA. These data revealed that the *trans* CMV not only truly functioned in *trans* via dCas9/csgRNA, but also synergistically interacted with dCas9-fused VP64. Subsequent transfections indicated that ZsGreen expression could also be highly activated by combination of dCas9-VP64, sCMV and other two csgRNAs, csgRNA1 and csgRNA3.

To further verify the function of CRISPR-assisted *trans* enhancer, we transfected six different cell lines with reporter construct and dCas9-VP64/csgRNA2-sCMV, dCas9/csgRNA2-sCMV, or dCas9-VP64/sgRNA (*Figure 2b*; *Figure 2—figure supplement 2–4*). The results revealed that dCas9-VP64/csgRNA2-sCMV always showed the highest gene activation efficiency in all cell lines. Additionally, dCas9/csgRNA2-sCMV always showed higher activity than dCas9-VP64/sgRNA. These results indicate that genes could be activated by the CRISPR-assisted *trans* enhancer. Importantly, the *trans* sCMV could synergistically function with dCas9-fused VP64 in gene activation.

## Comparison of *trans* CMV enhancer with VPR

Having shown that VPR is a more potent transcriptional activation domain than VP64, we next compared the *trans* enhancer with VPR. 293T and HepG2 cells were, respectively, transfected with reporter construct and dCas9-VP64/csgRNA, dCas9-VPR/csgRNA, or dCas9-VP64/csgRNA-sCMV (*Figure 2c*). The results showed that dCas9-VPR/csgRNA had better activity than dCas9-VP64/csgRNA as previously reported. However, the dCas9-VP64/csgRNA-sCMV always showed significantly higher activity than dCas9-VPR/csgRNA.

## Activation of endogenous genes by *trans* enhancer

To further evaluate the activity of CRISPR-assisted *trans* enhancer, we activated endogenous genes with *trans* sCMV. csgRNAs targeting ten different genes was designed and their linear expression vectors were produced. Seven different cell lines were transfected with dCas9-VP64/csgRNA2-sCMV, dCas9-VP64/sgRNA and dCas9/csgRNA2-sCMV (*Figure 3—figure supplement 1*). The quantitative PCR (qPCR) detection of gene expression revealed that almost all genes were most significantly activated by dCas9-VP64/csgRNA2-sCMV in all cells. Moreover, most genes were more significantly activated by dCas9/csgRNA-sCMV than dCas9-VP64/sgRNA in all cells. These results suggest that the CRISPR-assisted *trans* enhancer could be used to activate variant endogenous genes in various cells. In addition, by activating the HNF4α gene in 293 T cells, we found that dCas9-VP64/csgRNA-sCMV had better activity than dCas9-VPR/csgRNA-sCMV in activating endogenous genes (*Figure 3—figure supplement 2*).

It has been reported that the cancer cells HepG2 and PANC1 can be differentiated into normal liver- and pancreas-like cells by exogenously expressing transcription factor HNF4α and E47. In the above assays, we found that the endogenous HNF4α and E47 genes were highly activated by the CRISPR-assisted *trans* enhancer in HepG2 and PANC1 cells (*Figure 3*). To further confirm the cellular effects of HNF4α and E47 activation, we detected expressions of other genes related to the differentiation of the two cancer cells (*Figure 4*). The results indicated that the genes related to stemness (CD133 and CD90) and pluripotency (Oct3/4, Sox2, Nanog, c-Myc, LIN28, and Klf4) were down-regulated, but those related to normal liver (GS, BR, ALDOB, CYP1a2, PEPCK, APOCIII, G-6-P, and HPD) and pancreas (MIST1, PRSS2, CELA3A, and CPA2) functions were highly up-regulated in HepG2 and PANC1 cells. Additionally, the cell cycle arrest-related gene p21 (HepG2 and PANC1) and TP53INP1 (PANC1) were highly up-regulated.

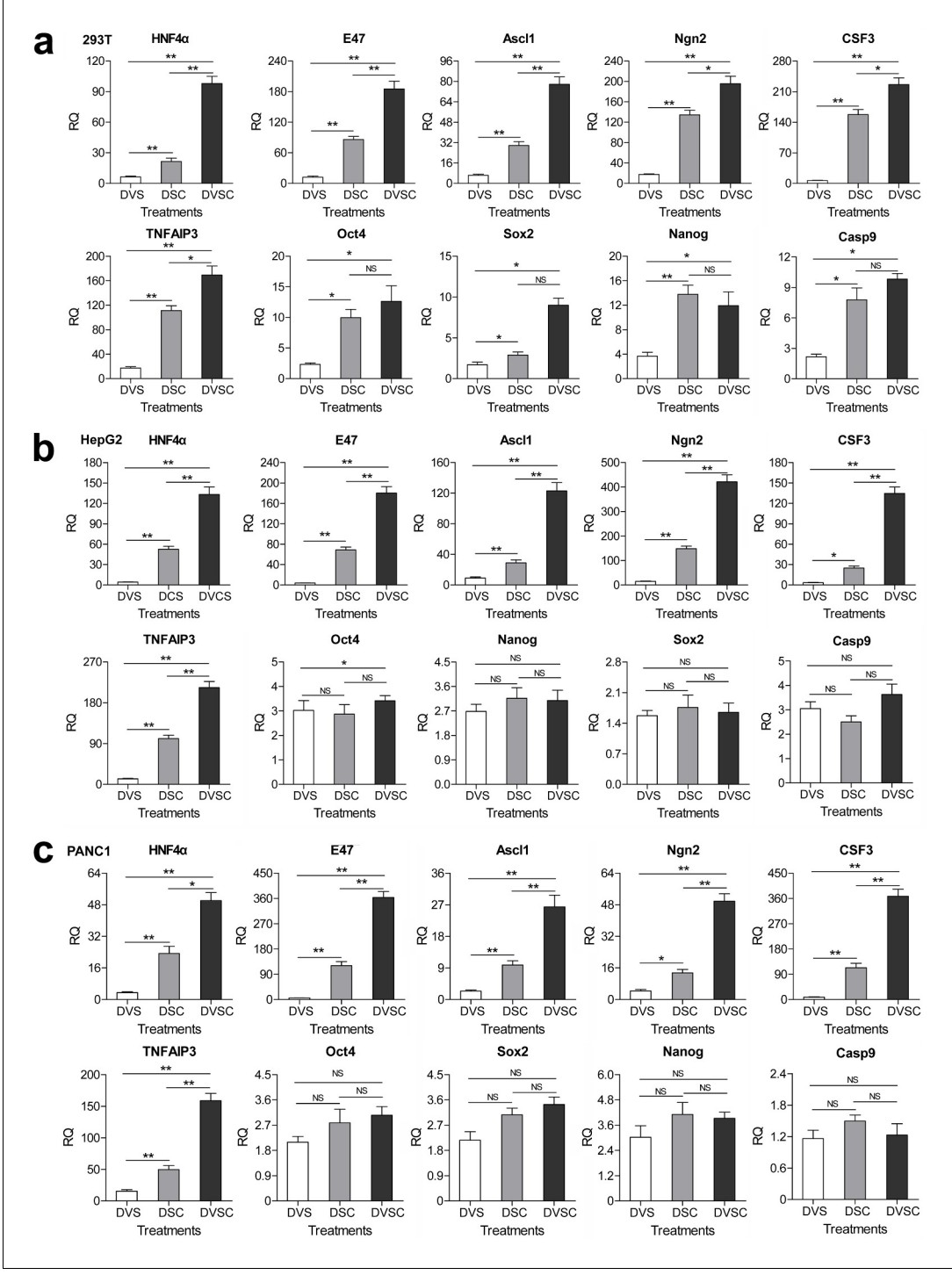

**Figure 3.** Transcriptional activation of endogenous genes by the CRISPR-assisted *trans* enhancer. 294T, HepG2 and PANC1 cells were transfected with three different transcriptional activation systems to activate expression of 10 endogenous genes. Gene transcription was detected by qPCR and the expression level is shown as the relative RNA expression fold to house-keeping gene GAPDH. Data are shown as mean ± SD, n = 3. The statistical difference was analyzed using the Student's *t* test. *, p<0.05; **, p<0.01; NS, no significant statistical difference. Transfection: DVS, dCas9-VP64/sgRNA; DSC, dCas9/csgRNA2-sCMV; DVSC, dCas9-VP64/csgRNA2-sCMV.
DOI: https://doi.org/10.7554/eLife.45973.008

The following figure supplements are available for figure 3:

**Figure supplement 1.** Transcriptional activation of endogenous genes by the CRISPR-assisted *trans* enhancer.
*Figure 3 continued on next page*

*Figure 3 continued*

DOI: https://doi.org/10.7554/eLife.45973.009

**Figure supplement 2.** Activation of endogenous HNF4α gene in 293 T cell with *trans* enhancers based on dCas9-VP64 and dCas9-VPR.

DOI: https://doi.org/10.7554/eLife.45973.010

## Activation of genes by other *trans* enhancers

To explore whether other enhancers could be also used as *trans* enhancers, we fabricated the blunt- and stick-end versions of two other widely used promoters EF1a and PGK (bEF1a, bPGK, sEF1a, and sPGK). 293 T cells were co-transfected with these *trans* enhancers and dCas9-VP64/csgRNA and reporter construct. The results indicated that the ZsGreen expression was also activated by the two *trans* enhancers; however, the activation levels were lower than that of sCMV (*Figure 5a*, *Figure 5—figure supplement 1*). All other transfections as controls did not activate ZsGreen expression (*Figure 5a*, *Figure 5—figure supplement 1*). The qPCR detection of HNF4α expression in the same transfected 293 T cells revealed that the endogenous HNF4α gene expression was also significantly activated by three stick-end *trans* enhancers, but not activated by all blunt-ended *trans* enhancers (*Figure 5b*). Subsequent HepG2 cell transfections with the same *trans* enhancers and dCas9-VP64/csgRNA indicated that the endogenous HNF4α gene expression could also be significantly activated by all stick-ended *trans* enhancers, but not activated by all blunt-ended *trans* enhancers (*Figure 5b*). These results indicate that the variant enhancers could be used as the CRISPR-assisted *trans* enhancer.

## Activation of genes by the GAL4/UAS-based *trans* enhancer

To further improve in vivo application of the CRISPR-assisted *trans* enhancer, we tried realizing the *trans* enhancer with the GAL4-UAS system. A dCas9-VP64-GAL4 expression vector and a UAS-CMV *trans* enhancer was constructed. Two forms of *trans* UAS-CMV enhancers, linear UAS-CMV (LUAS-CMV) and circular UAS-CMV (CUAS-CMV), were expected to be recruited to the target gene by the dCas9-VP64-fused GAL4 (*Figure 6a*). By transfecting 293 T cells with dCas9-VP64-GAL4/sgRNA-LUAS-CMV/CUAS-CMV and reporter construct, the ZsGreen expression of the exogenous reporter gene was significantly activated by both LUAS-CMV and CUAS-CMV, but not activated by all transfections as controls (*Figure 6b*, *Figure 6—figure supplement 1*). By transfecting 293T and HepG2 cells with dCas9-VP64-GAL4/sgRNA-LUAS-CMV/CUAS-CMV, the expression of endogenous HNF4α gene was highly activated in the two cells (*Figure 6c*). More importantly, both *trans* LUAS-CMV and CUAS-CMV enhancers showed significantly higher activity than the *trans* sCMV (*Figure 6c*). In contrast, all transfections as controls did not activate the expression of endogenous HNF4α gene in the two cells (*Figure 6c*). These results reveal that the CRISPR-assisted *trans* enhancer could be better realized with the GAL4-UAS system.

## Discussion

In this study, we developed a new dCas9-based gene activation strategy, the CRISPR-assisted *trans* enhancer, in which a *trans* enhancer could be recruited to target promoters by dCas9-VP64/csgRNA or dCas9-VP64-GAL4/sgRNA. The results revealed that expression of variant exogenous and endogenous genes could be highly activated by CRISPR-assisted *trans* enhancers in various mammalian cells, more efficiently than with current widely used dCas9-VP64 and dCas9-VPR. This strategy has unique advantages over the current dCas9-based gene activation systems.

First, only one csgRNA was used in activating all target genes in various cells with the CRISPR-assisted *trans* enhancer. However, in current dCas9-based gene activations, multiple sgRNAs are used. In general, three or more sgRNAs are used to activate a gene of interest (*Cheng et al., 2013*; *Maeder et al., 2013*; *Mali et al., 2013a*; *Perez-Pinera et al., 2013*). In many assays with various numbers of sgRNAs, one sgRNA often produced very low or undetectable expression. Using multiple sgRNAs, each sgRNA has to be independently transcribed by a long U6 promoter. Second, csgRNA is the simplest sgRNA used in dCas9-based gene activators, which only extended a 24 bp short sequence at the 3′ end of normal sgRNA. However, current dCas9/sgRNA activators often use

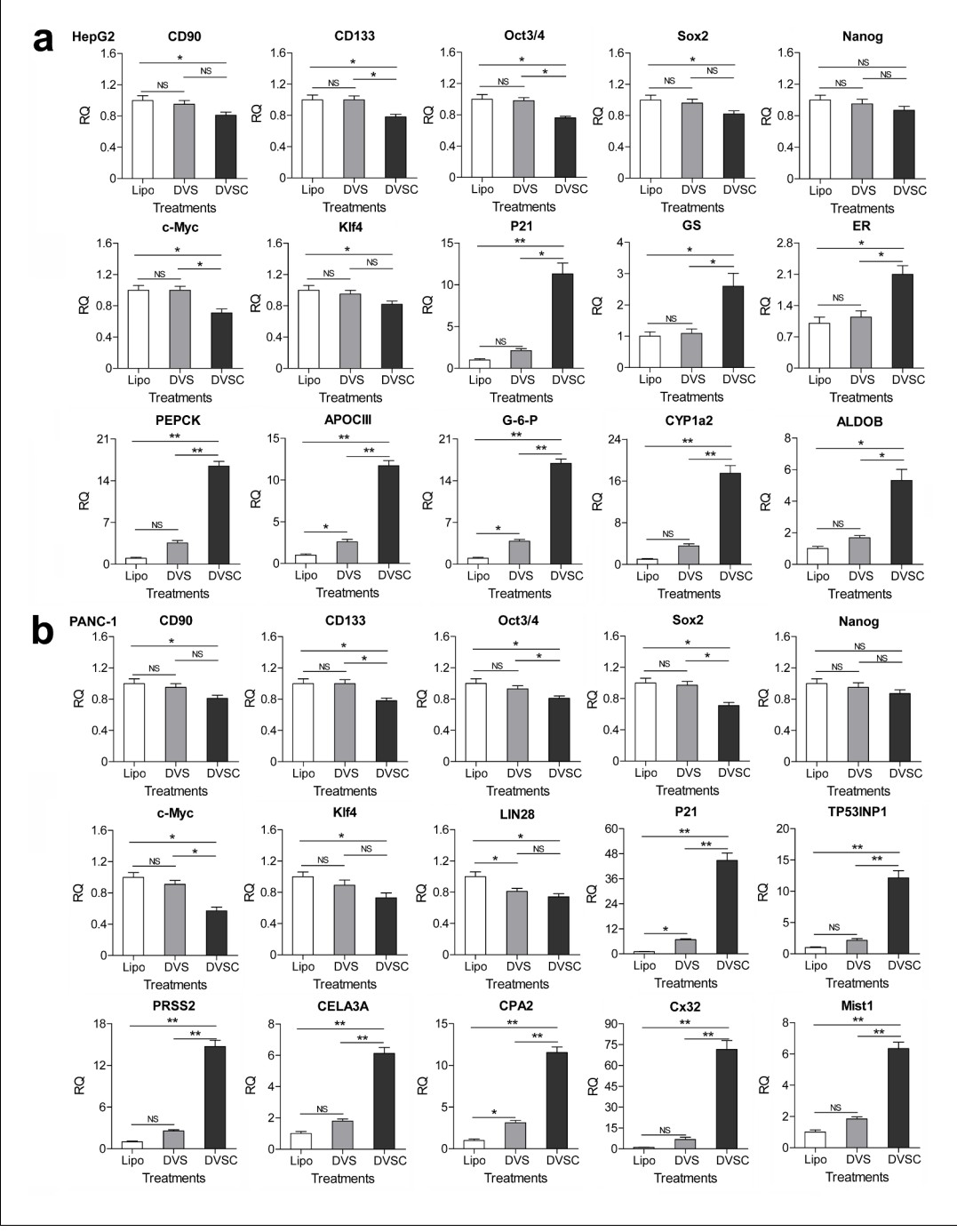

**Figure 4.** Changes of gene expression in the HNF4α-activated HepG2 cells and E47-activated PANC-1 cells. (**a** and **b**) Changes of gene expression in the HNF4α-activated HepG2 cells (**a**) and E47-activated PANC-1 cells (**b**). The gene transcription was detected by qPCR and the expression level is shown as the relative RNA expression fold to house-keeping gene GAPDH. Data are shown as mean ± SD, n = 3. The statistical difference was analyzed by Student's *t* test. *, p<0.05; **, p<0.01; NS, no significant statistical difference. Transfection: Lipo, lipofectin; DVS, dCas9-VP64/sgRNA; DVSC, dCas9-VP64/csgRNA2-sCMV.

DOI: https://doi.org/10.7554/eLife.45973.011

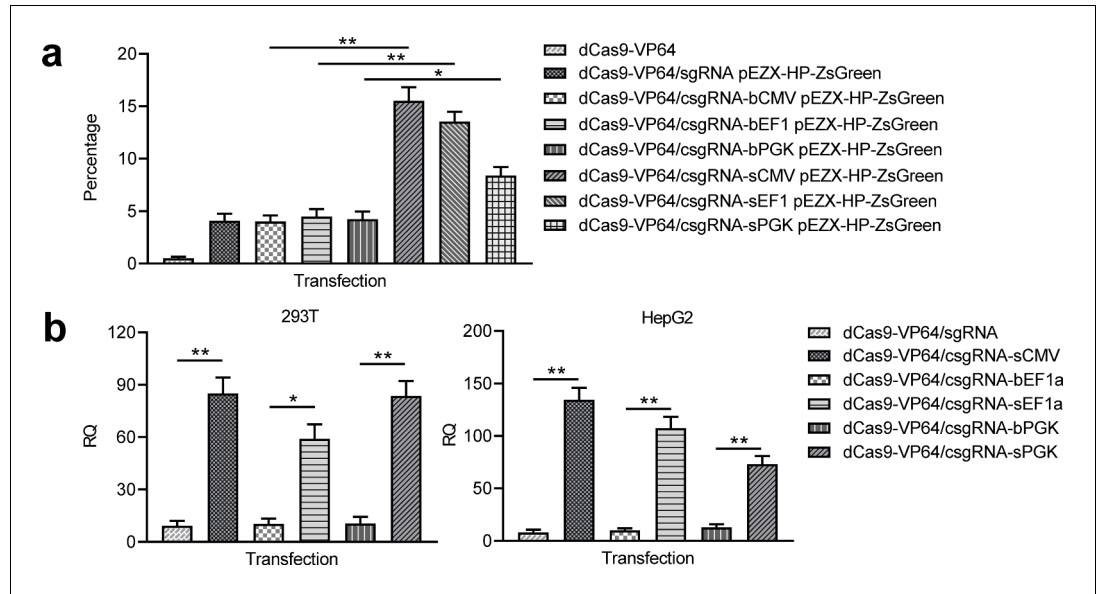

**Figure 5.** Activation of exogenous and endogenous genes with other *trans* enhancers. (**a**) Activation of exogenous reporter gene ZsGreen. The fluorescence intensity of cells was analyzed with flow cytometry. (**b**) Activation of endogenous HNF4α gene in 293T and HepG2 cells with two new *trans* enhancers, sEF1a and sPGK. The sCMV was used as a positive control for comparison. The blunt-end *trans* enhancers (bEF1a and bPGK) were also used as controls. Data are shown as mean ± SD, n = 3. The statistical difference was analyzed using Student's *t* test. *, p<0.05; **, p<0.01.

DOI: https://doi.org/10.7554/eLife.45973.012

The following figure supplement is available for figure 5:

**Figure supplement 1.** Activation of exogenous and endogenous genes with other CRISPR-assisted *trans* enhancers.

DOI: https://doi.org/10.7554/eLife.45973.013

---

long complex chimeric sgRNAs that harbor multiple tandem aptamers of various RNA-binding proteins, such as SAM sgRNA (MS2) (*Konermann et al., 2015*; *Liao et al., 2017*), Casilio sgRNA (Pumilio/FBF) (*Cheng et al., 2016*), and scaffold RNAs (MCP, PCP, and Com) (*Zalatan et al., 2015*).

The capture sequences of csgRNA can be easily designed. We originally designed three different capture sequences. All functioned in the CRISPR-assisted *trans* enhancer; however, csgRNA2 showed the best performance. The capture sequences were artificially designed short sequences, they have no complementary sequences in human cells, which is important for their specific annealing with sCMV at high efficiency. This study demonstrated that sCMV could be efficiently captured by csgRNA in the nucleus of human cells. To our knowledge, this is the first report of a gene being activated by an artificial DNA in *trans*.

In this study, we realized the CRISPR-assisted *trans* enhancer with two forms: csgRNA-sCMV and GAL4-UAS. Two forms can be easily used to activate genes in in vitro cultivated cells. As to the in vivo applications, the csgRNA-sCMV-based *trans* enhancer can be used via nanoparticle gene carriers, while the GAL4-UAS-based *trans* enhancer can be easily transferred by virus vectors such as AAV, with AAV already being approved for use as a gene vector in clinics. We found that the GAL4-UAS-based *trans* enhancer had better performance than the csgRNA-sCMV-based *trans* enhancer, especially the linear UAS-CMV. In in vivo applications, the linear UAS-CMV can be easily transferred by AAV vector.

As a typical application, dCas9-based transcriptional activators are used to reprogram cells in vitro and in vivo for biomedical applications by activating endogenous genes. For example, fibroblasts were reprogramed into induced pluripotent stem (iPS) cells by endogenous activation of the Oct4 and Sox2 genes with dCas9-SunTag-VP64 (*Liu et al., 2018*). Mouse embryonic fibroblasts were converted into neuronal cells by endogenous activation of the Brn2, Ascl1, and Myt1l genes with [VP64]dCas9[VP64] (*Black et al., 2016*). In vivo target genes were activated by MPH to ameliorate

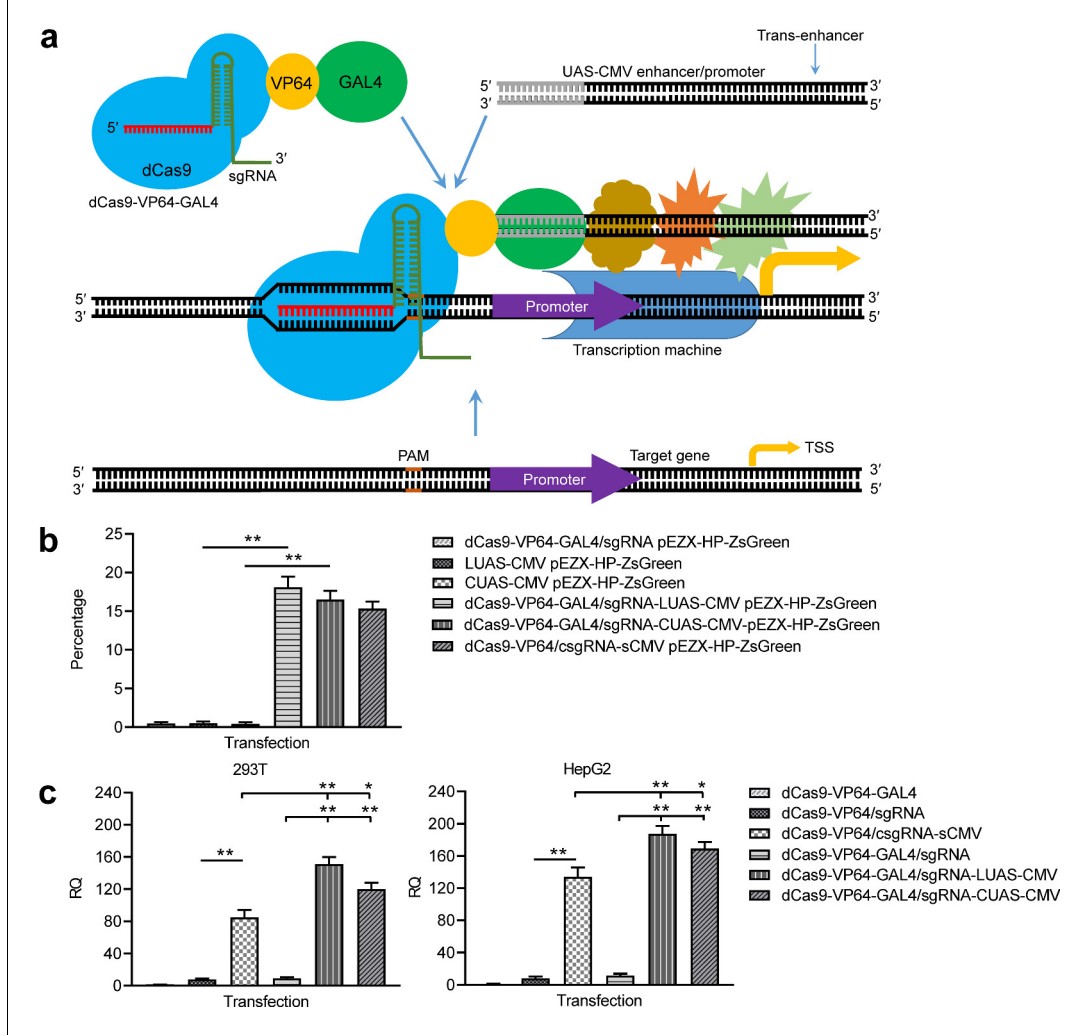

**Figure 6.** Activation of exogenous and endogenous genes with the GAL4-UAS-based *trans* enhancer. (**a**) Schematic show of gene activation using the GAL4-UAS-based CRISPR-assisted *trans* enhancer. (**b**) Activation of exogenous reporter gene ZsGreen. The fluorescence intensity of cells was analyzed with flow cytometry. (**c**) Activation of endogenous HNF4α gene in 293T and HepG2 cells with the GAL4-UAS-based CRISPR-assisted *trans* enhancer. The sCMV was used as a positive control for comparison. Three other transfections were used as controls. LUAS-CMV, linear UAS-CMV; CUAS-CMV, circular UAS-CMV. Data are shown as mean ± SD, n = 3. The statistical difference was analyzed by Student's t-test. *, p<0.05; **, p<0.01.
DOI: https://doi.org/10.7554/eLife.45973.014

The following figure supplement is available for figure 6:

**Figure supplement 1.** Activation of exogenous and endogenous genes with CRISPR-assisted *trans* enhancer using the GAL4-UAS system.
DOI: https://doi.org/10.7554/eLife.45973.015

disease phenotypes in mouse models of type I diabetes, acute kidney injury, and muscular dystrophy (*Liao et al., 2017*). Brain astrocytes were converted into functional neurons in vivo by activating the Ascl1, Neurog2 and Neurod1 genes with SPH (*Zhou et al., 2018*). These studies make CRISPR therapies the grade not the cut (*Burgess, 2018*).

In this study, we selected 10 endogenous genes to be activated by the CRISPR-assisted *trans* enhancer. Most of these genes code transcription factors, including HNF4α, E47, Ascl1, Ngn2, Sox2, Oct4, and Nanog. Ascl1, Ngn2, and Sox2 are used to directly reprogram fibroblasts into nerve cells (*Zhao et al., 2015*). Oct4, Sox2, and Nanog are widely used to reprogram fibroblasts into iPS cells (*Takahashi et al., 2007*; *Takahashi and Yamanaka, 2016*; *Yu et al., 2007*). HNF4α and E47 are used to differentiate liver and panaceas cancer cells into normal cells (*Kim et al., 2015*; *Yin et al., 2008*). TNFAIP3 is a well-known natural NF-κB inhibitor (*Cooper et al., 1996*), having the potential to treat NF-κB-overactivated diseases such as inflammation and cancers. Caspase9 is a key gene

making cell apoptosis (*Li et al., 2017*). CSF3 codes granulocyte-colony stimulating factor (G-CSF), a glycoprotein that stimulates the bone marrow to produce granulocytes and stem cells and release them into bloodstream (*Cetean et al., 2015*), and is widely used in chemotherapy to enhance the immunity of cancer patients. We selected these genes for exploring the future in vitro and in vivo applications of the CRISPR-assisted *trans* enhancer.

## Materials and methods

### Vector construction

A lac operon fragment was amplified from pEASY-Blunt-simple (Transgen) using primers Lac-px-F and Lac-px-R. The product was ligated into px458 (Addgene) to construct px458-lac using BbsI (NEB) and BsaI (NEB). The U6-sgRNA-lac fragment was amplified from px458-lac using primers U6-F and U6-R/U6-1-R/U6-2-R/U6-3-R. The products were cloned into the pEASY-Blunt-simple to produce pEASY-sgRNA and pEASY-csgRNA (*Supplementary file 2*), which were used to construct particular sgRNA/csgRNA expressing plasmids. The sgRNAs targeting genes of interest were designed by CHOPCHOP. Chemically synthesized complementary oligonucleotides containing sgRNA/csgRNA targets were annealed and ligated into pEASY-sgRNA/pEASY-csgRNA. The ligation reaction consisted of 10 U BbsI (NEB), 120 U T4 DNA ligase (NEB), $1 \times$ T4 DNA ligase buffer, 0.1 mg/mL bovine serum albumin, and 50 ng plasmid pEASY-csgRNA. The reaction was run as follows: 10 rounds of 37 °C5 min and 16°C 10 min, 37°C 30 min, and 80°C 5 min. The reaction was then used to transfect DH5α competent cells. The white clones were screened by blue-white screening on LB agar plates with 100 µg/mL ampicillin, 40 µL of 20 mg/mL X-gal, and 40 µL of 0.1 M IPTG. The vectors were validated by sequencing, then the linear sgRNA/csgRNA expression vectors were amplified from the validated pEASY-sgRNA/pEASY-csgRNAs using primers U6-F and U6-R/U6-1-R/U6-2-R/U6-3-R. The primer U6-R was used to amplify the normal sgRNA expression template (named as U6-sgRNA) from pEASY-sgRNA. The primer U6-1/2/3 R was used to amplify the csgRNA expression template (named as U6-csgRNA) from pEASY-csgRNA. The PCR products were purified with PCR clean kit (Axygen) and used to transfect cells as sgRNA/csgRNA expression vector.

The CMV enhancer fragment was amplified from pEGFP-N1 using primers CMV-F and CMV-1-R/CMV-2-R/CMV-3-R. The PCR products were purified with PCR clean kit and used as linear blunt-end CMV (bCMV). To prepare stick-end CMV (sCMV), the PCR products were firstly digested with Nt. BbvCI and then added with complementary oligonucleotide CS-1-R/CS-2-R/CS-3-R. The PCR products were denatured at 85°C for 10 min and then naturally cooled to room temperature. The PCR products was purified with PCR clean kit and used as linear sCMV. The blunt-end EF1α/PGK promoters were all amplified from pEF1a-FB-dCas9-puro (Addgene) using primers EF1-a-F/R and PGK-F/R. The stick-end EF1α/PGK promoters were similarly prepared by treating blunt-end EF1α/PGK promoters.

A 1000 bp HNF4α promoter sequence was amplified from the genomic DNA of HepG2 cells using primers HNF4α-P-F and HNF4α-P-R. The amplified promoter fragment was ligated into pEZX-ZsGreen, producing an HNF4α promoter reporter construct pEZX-HP-ZsGreen. The VP64 sequence was deleted from pcDNA-dCas9-VP64 (Addgene) to construct pcDNA-dCas9. The VPR sequence was cloned into pcDNA-dCas9 to construct pcDNA-dCas9-VPR.

The GAL4 fragment was amplified from pGBKT7 (MiaoLing Plasmid Sharing Platform) using primers GAL4-BsiEW-F and GAL4-BspE-R, and was then ligated into pcDNA-dCas9-VP64 using BsiWI and BspEI to prepare pcDNA-dCas9-VP64-GAL4. A chemically synthesized 5 × UAS fragment was ligated into pEASY-Blunt using BglII and HindIII to obtain pEASY-Blunt-UAS. A CMV fragment was amplified from pEGFP-C1 using primers UAS-CMV-Bgl-F and UAS-CMV-Hind-R. The CMV fragment was then ligated into pEASY-Blunt-UAS to obtain pEASY-Blunt-UAS-CMV, which was used as circular UAS-CMV (CUAS-CMV). The linear UAS-CMV (LUAS-CMV) fragment was amplified from pEASY-Blunt-UAS-CMV using primers CMV-UAS-Bgl-F and UAS-CMV-R.

The sequences of all PCR primers used in the above vector construction are shown in the *Supplementary file 1*–Table 1. The chemically synthesized complementary oligonucleotides used to construct pEASY-sgRNA/pEASY-csgRNA are shown in the *Supplementary file 1*–Table 2. The functional sequences of all linear and plasmid vectors are provided as *Supplementary file 3*.

## DNA cutting with Cas9-csgRNA

A sgRNA targeting the HNF4α promoter sequence was selected. The sgRNAs were prepared by in vitro transcription using T7 RNA polymerase (NEB). The sgRNA transcription template was amplified from pEASY-csgRNA using primers HNF4α-T7-F and U6-R/U6-1-R/U6-2-R/U6-3-R. A normal sgRNA (HNF4α-sgRNA) and three csgRNAs (HNF4α-csgRNAs) were prepared. A 732 bp HNF4α promoter fragment was amplified from pEZX-HP-ZsGreen using primers HNF4α-sP-F and HNF4α-sP-R. The sequences of PCR primers are shown in the *Supplementary file 1*-Table 1. The Cas9 digestion reaction (30 μL) consisted of 1 × Cas9 Nuclease Reaction Buffer, 1 μM Cas9 Nuclease (NEB), and 300 nM HNF4α-sgRNA or HNF4α-csgRNA. The reaction was incubated at 25°C for 10 min. Then 400 ng of purified 732 bp HNF4α promoter fragment was added to the reaction and incubated at 37°C for 15 min. Finally, the Cas9 nuclease was inactivated at 65°C for 10 min. The reaction was run with 1.5% agarose gel electrophoresis.

## Cell lines

All cells including 293T, HepG2, PANC1, A549, HeLa, SKOV3, and HT29 were obtained from the Shanghai Institutes for Biological Sciences, Chinese Academy of Sciences. The identity was authenticated by STR profiling. Mycoplasma contamination testing was performed and no mycoplasma contamination was ensured.

## Cell culture and transfection

Cells were cultured in Dulbecco's Modified Eagle Medium (DMEM) or Roswell Park Memorial Institute (RPMI) 1640 medium supplemented with 10% FBS, 100 units/mL penicillin, and 100 μg/mL streptomycin. Cells at >70% confluence in each well of a 12-well plate were transfected with various combinations (see figures) of linear or plasmid vectors, including pcDNA-dCas9, pcDNA-dCas9-VP64, pcDNA-dCas9-VPR, U6-sgRNA, U6-csgRNA, sCMV, bCMV, and pEZX-HP-ZsGreen, using Lipofectamine 2000 (ThermoFisher Scientific) according to the manufacturer's instructions. The transfected cells were incubated at 37°C and 5% $CO_2$ for 36 h. Cells were then imaged with a fluorescence microscope (Olympus) at 200 × magnification.

## Flow cytometry

The fluorescence intensity of cells was quantified with flow cytometry (Calibur). Ten-thousand cells were measured for each transfection. Flow cytometry data analysis and figure preparation were done using BD software.

## Quantitative PCR

The total RNA was extracted from cells using TRIzol (Invitrogen). The complementary DNA (cDNA) was synthesized with 3 μg of total RNA using the Hifair III SuperMix (Yeasen). Gene transcription was detected with quantitative PCR (qPCR) using the Hieff qPCR SYBR Green Master Mix (Yeasen) according to the manufacturer's instructions. GAPDH was used as an internal reference to analyze the relative mRNA expression of different genes. The sequences of PCR primers are shown in the *Supplementary file 1*-Table 3 and 4. The qPCR programs were run on StepOne Plus (Applied Biosystems). Each qPCR detection was performed in at least three technical replicates. Melting curve analysis was performed. Data analysis was performed using the Applied Biosystems StepOne software v2.3, and $C_t$ values were normalized with that of GAPDH. The relative expression level of target mRNAs was calculated as relative quantity (RQ) according to the equation: RQ = $2^{-\Delta\Delta Ct}$.

## Statistical analyses

Each cell transfection for detecting gene expression activation by *trans* enhancer was performed in three biological replicates. In each biological replicate, at least three technical replicates (three replicate wells) were performed. In qPCR detection of gene expression, the mean RQ value of technical replicates was used as the RQ value of one biological replicate. The mean RQ value of three biological replicates was used to calculate the final mean and standard deviation (SD). Data were analyzed by Student's *t* test when comparing two groups. Data are shown as mean ± SD, and differences were considered significant at p<0.05.

# Additional information

## Funding

| Funder | Grant reference number | Author |
|---|---|---|
| National Natural Science Foundation of China | 61571119 | Jinke Wang |

The funders had no role in study design, data collection and interpretation, or the decision to submit the work for publication.

## Author contributions

Xinhui Xu, Methodology, Writing—original draft, Writing—review and editing, Performed the csgRNA-sCMV-based experiments; Jinliang Gao, Methodology, Writing—review and editing, Performed the GAL4-UAS-based experiments; Wei Dai, Methodology, Writing—review and editing, Helped to prepare solutions and culture cells; Danyang Wang, Methodology, Writing—review and editing, Helped to construct vectors; Jian Wu, Data curation, Software, Validation, Writing—review and editing, Helped to analyze data; Jinke Wang, Conceptualization, Supervision, Funding acquisition, Writing—original draft, Writing—review and editing, Conceptualized, designed and supervised the research, Wrote the manuscript and provided financial support for the project

## Author ORCIDs

Jinke Wang http://orcid.org/0000-0002-3352-4690

## Decision letter and Author response

Decision letter https://doi.org/10.7554/eLife.45973.021
Author response https://doi.org/10.7554/eLife.45973.022

# Additional files

## Supplementary files

• Supplementary file 1. Four tables showing primers and oligos.
DOI: https://doi.org/10.7554/eLife.45973.016

• Supplementary file 2. Schematic show of construction of sgRNA vectors for blue-white screening.
DOI: https://doi.org/10.7554/eLife.45973.017

• Supplementary file 3. Sequences of vectors, templates, sgRNA, csgRNA, and *trans* enhancers.
DOI: https://doi.org/10.7554/eLife.45973.018

• Transparent reporting form
DOI: https://doi.org/10.7554/eLife.45973.019

## Data availability

All data generated or analysed during this study are included in the manuscript and supporting files.

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
