## [Decision Letter]

[Editors’ note: a previous version of this study was rejected after peer review, but the authors submitted for reconsideration. The first decision letter after peer review is shown below.]

Thank you for submitting your work entitled "Gene activation by a CRISPR-assistant *trans* enhancer" for consideration by *eLife*. Your article has been reviewed by three peer reviewers, including Irwin Davidson as the Reviewing Editor and Reviewer #1, and the evaluation has been overseen by a Reviewing Editor and a Senior Editor.

Our decision has been reached after consultation between the reviewers. Based on these discussions and the individual reviews below, we regret to inform you that your work will not be considered further for publication in *eLife*.

While the referees found the concept of tethering an enhancer element in trans as a novel and innovative approach to activate expression of chosen endogenous cellular genes, they were unanimously concerned by the lack of information on the reproducibility of the data, the absence of appropriate statistical tests and important missing controls. The presentation of the data made it difficult to assess the efficiency of the system and the many of the experiments will have to be repeated again and the data presented in a much more rigorous fashion. The referees would encourage the authors to take note of the many suggestions and comments that would have to be addressed in any future manuscript.

*Reviewer #1:*

This paper describes a novel system to tether an CMV enhancer element in trans to selected regions of the genome using CRISPR/CAS9 allowing the selective activation of chosen target genes. They show that activation by this method is more efficient that CAS9-VP64 and that the two systems can be used cooperatively.

Major issues:

To demonstrate the specificity of the technique, the authors should perform experiments with a version of the CMV enhancer in which key bases have been mutated to impair the binding of transcription factors. This would demonstrate the importance of using a functional enhancer. The authors may also consider experiments using cell-specific enhancers and demonstrating that the system can used in a cell-type specific manner. This is not essential for the message of the paper but may give added value to the study.

In Figure 2 and Figure 3 as well as Figure 2—figure supplement 2, Figure 2—figure supplement 3, Figure 2—figure supplement 4 the results of the ZsGreen assays are absolutely not clear. What are we supposed to compare, how by looking at the figures can we assess the activation in the different conditions? Also, in the text, there are no precise figures given concerning the fold activation in each condition. In Figure 4A, there are only 2 transfections so there are no robust statistics concerning activation. The reasons for assaying ZsGreen rather measuring RNA output directly are not well justified. All of these figures have to be modified such that the reader can readily assess the differences in each experimental condition. RNA should be measured by RT-qPCR and fold changes and statistics from replicate experiments should be provided.

In Figure 5 and Figure 6 the number of biological replicates and the statistical tests used should both be indicated. The text describing these figures is somewhat oversimplified as they claim that activation of HNF4A results in strong induction of liver function genes. In fact, some are strongly induced, but for others the effect is very low and this is not recognised in the text. Statistics should be shown for the cell cycle analyses in Figure 6C number of replicates and statistical test used.

Figure S9. There is no obvious difference in the number of cells in the different conditions. This must be quantitated and analysed to provide reliable statistics.

In summary, while this paper describes an interesting and novel technique, the data as they are presented require additional replicates, statistical test and clearer presentation to be acceptable.

Additional data files and statistical comments:

The number of replicates and the statistical tests used are not indicated and more rigorous analyses have to be performed and present in any revised version. The present version is way below acceptable standards.

*Reviewer #2:*

Here, Xu et al., describe a transcriptional activation strategy using dCas9-VP64 together with a modified gRNA scaffold (csgRNA) that can anneal to a CMV enhancer DNA sequence (sCMV) provided in trans. They show that dCas9/csgRNA/sCMV (in the absence of a transactivation domain) is capable of inducing transcription of a reporter construct and that the transcriptional activity of the complex is enhanced by the addition of a transactivation domain (VP64) to dCas9 (dCas9-VP64). They also show that transfecting dCas9-VP64/csgRNA/sCMV is capable of inducing transcription of the reporter in 7 different cell lines. They also claim that dCas9-VP64/csgRNA/sCMV is a little bit more efficient that dCas9-VPR in inducing transcription. Finally, they show that dCas9-VP64/csgRNA/sCMV can induce transcription of endogenous genes in two different cancer cell lines and that this can trigger differentiation.

Although the strategy proposed here is original, the sCMV component needs to be provided in trans (co-transfected), therefore application of this system is limited to cells in culture and would be very challenging to use this strategy in vivo.

A major flaw of this study is that transfection efficiency cannot be determined. It is impossible to know the percentage of cells that are expressing dCas9-VP64 and the proportion of these in which transcriptional activation has occurred. Therefore, the transcriptional effect observed is either under- or over-estimated.

There are many variables in the experiment, as 4 constructs need to be co-transfected (reporter, dCas9-VP64, csgRNA and sCMV). The experiments should have been done in a cell line stably transfected with the reporter construct and using a single plasmid co-expressing the csgRNA together with mCherry-dCas9-VP64 (or mCherry-dCas9-VPR). This would not only reduce the number of variables but would allow the authors to: (1) Determine the transfection efficiency and the frequency of mCherry+ cells (expressing dCas9-VP64) in which transcriptional activation has occurred (ZsGreen+) and (2) Determine the transactivation level by scoring the MFI of ZsGreen in the mCherry+ ZsGreen+ population.

The FACS data should be presented as dot plots (FSC vs. ZsGreen). This allows to visualize the ZsGreen+ population, which is not clear at all (given the low percentage of ZsGreen+ cells) from the histograms.

The manuscript is poorly written and is full of strange words and expressions and should be proof-read by an English-speaking person.

Figure legends do not describe the figures and should be re-written. Legend for Figure 1B is missing.

On all FACS data shown, the gating for ZsGreen is not the same throughout (i.e. compare panels in Figure 2).

It is not clear how the MFI intensity in Figure 4 was determined. Therefore, it is impossible to assess the validity of the comparisons.

In Figure 1B, the percentage of cleavage should be displayed, as it is clear that the csgRNA is not as efficient as the unmodified scaffold. Also, the input is missing.

In Figure 5, a negative control without the csgRNA is missing. The relative mRNA induction should be calculated relative to this control, as it would directly show the fold induction triggered by dCas9-VP64/csgRNA/sCMV. Also, the y axis should be labelled relative mRNA (and not RNA) expression.

In the qPCR data shown in Figure 6A and 6B, it is not indicated what was used to determine the relative expression. Same as in Figure 5, the negative control (without the csgRNA) is missing and should be performed in order to determine the relative expression of each gene. Also, the expression level of the target gene should be shown.

For clarity, a diagram of the reporter construct should be shown.

Additional data files and statistical comments:

Throughout the manuscript, it is not clear how many times the experiments were done. For example, the qPCR data shown in Figure 5, do the error bars correspond to technical or biological replicates? Is a representative experiment shown? Also, in several figures it is not indicated which samples are compared when the * is shown for statistical significance.

*Reviewer #3:*

In their study, Xu et al., developed a novel approach using CRISPR/dCas9 to activate gene expression. They modified the guide RNA scaffold as such that a CMV enhancer DNA fragment was recruited to the target gene by the dCas9-VP64/sgRNA complex. The authors argue that previously published approaches have limitation regarding the strength of transcriptional activation and their approach is superior in this regard.

The approach by Xu et al., is novel and innovative having potential beyond the use case presented in the current study. Based on the data presented I have the following comments/questions.

1) The authors initially validate their approach (Figure 2) in 293T cells by co-transfecting an exogenous GFP-based reporter construct. Transient transfection efficiencies of 293T cells are known to be extremely high. Why does e.g. dCas9-VP64/csgRNA2-sCMV achieve only 11% of GFP+ cells, assuming that >90% of 293T are transfected?

2) Regarding Figure 5 and along the line of my previous comment. How do we have to interpret the fold-changes in expression of endogenous genes having seen that in Figure 2 only a comparably small proportion of cells activated the reporter construct? Based on the methods description, all endogenous gene expression analyses were done using the bulk cell cultures. Are the changes in gene expression seen in the bulk culture caused by extremely high transactivation in a small fraction of cells or by homogenous induction across all cells? At least the first scenario is suggested by the pilot experiments shown in Figure 2. Thus, single cell-based approaches could clarify this for the endogenous target genes, e.g. immunofluorescence/FACS for the proteins encoded by the transactivated target genes.

3) If transcriptional activation is achieved only in a fraction (approximately 10%) of the cells despite high transfection efficiencies, then the approach by Xu et al., would have some limitations, at least for bulk cell culture analyses, in comparison to SAM and other methods, which can be also efficiently delivered to target cells by viruses. Where do the authors then see the advantage of their approach, as the requirement for transient transfection remains a drawback, in particular in cell models that are difficult to transfect?

[Editors’ note: what now follows is the decision letter after the authors submitted for further consideration.]

Thank you for resubmitting your work entitled "Gene activation by a CRISPR-assisted trans enhancer" for further consideration at *eLife*. Your revised article has been favorably evaluated by Kevin Struhl (Senior Editor), a Reviewing Editor, and two reviewers.

The manuscript has been improved but there are some remaining issues that need to be addressed before acceptance, as outlined below:

In this new version, the authors addressed many of the issues raised on the previous versions. They have provided data on the reproducibility of the results with the appropriate statistics. They have used alternative regulatory elements in place of CMV and they have developed an alternative strategy for recruiting the trans-acting element using the GAL4 DNA binding domain. These new innovations have strengthened the study. However, the paper is now too long with too many figures. The paper would be more appropriate if it was shortened to a Tools and Resources article rather than a full manuscript. This can be done by reorganizing the figures. For example, rather than showing all of the fluorescence images and sorting results, this could all be shown in graphical form as is already the case for Figures 8C, D and E and Figure 9C, D and E. This would save a lot of space and condense all the information into a single figure, showing perhaps a single representative sorting experiment and cell images as supplemental data. Also Figure 5 could be shortened to show only 1 or 2 cell types with the rest shown as supplemental data. I also propose that the data on cell cycle changes and migration be removed to shorten the text. This data is not central to the main message of the paper. On the other hand, supplemental figure 11 should be shown in the main figures possibly as an addition to Figure 1.

Finally, the writing is poor and the meaning of what is written is in several places very difficult to understand. Mitigating this will require a major effort.

The authors are invited to submit a shortened revised version based on the above suggestions.

---

## [Author Response]

[Editors’ note: the author responses to the first round of peer review follow.]Reviewer #1:[…] To demonstrate the specificity of the technique, the authors should perform experiments with a version of the CMV enhancer in which key bases have been mutated to impair the binding of transcription factors. This would demonstrate the importance of using a functional enhancer. The authors may also consider experiments using cell-specific enhancers and demonstrating that the system can used in a cell-type specific manner. This is not essential for the message of the paper but may give added value to the study.

In order to obtain the significant gene activation, we used a widely used mammalian enhancer/promoter, CMV enhancer/promoter. This was described in detail in the Introduction:

“Almost three decades ago, the human cytomegalovirus (CMV) enhancer/promoter (referred to as CMV enhancer hereafter) was found as a natural mammalian promoter with high transcriptional activity (Boshart et al. 1985). The late studies gradually found that the CMV enhancer is the known strongest promoter in various mammalian cells (Boshart et al., 1985; Foecking and Hofstetter, 1986; Ho et al., 2015; Kim et al., 1990). Therefore, this enhancer has been widely used to drive the ectopic expression of various genes in wide range of mammalian cells. For example, the CMV enhancer is also used to drive the ectopic expressions of exogenous genes in broad tissues in transgenic animals (Furth et al., 1991; Schmidt et al., 1990), protein production by gene engineering, and human gene therapy. We have recently further improved the transcriptional activity of the CMV enhancer by changing the natural NF-κB binding sites in this enhancer into artificially selected NF-κB binding sequences with high binding affinity (Wang et al., 2018). Therefore, we conceived that an unique architecture may be constructed to further improve dCas9-based activators by using the CMV enhancer.”

CMV enhancer/promoter is a known strong natural mammalian enhancer/promoter. Therefore, it was widely employed in researches of biological sciences and biomedicine. Its strongest transcriptional activity was found to be dependent on many binding sites of multiple transcription factors. For example, there are four binding sites of NF-κB. We have recently published a paper about CMV enhancer/promoter in which we discussed in detail the CMV enhancer/promoter and further improved the transcriptional activity of the CMV enhancer by changing the natural NF-κB binding sites in this enhancer into artificially selected NF-κB binding sequences with high binding affinity (Wang et al., 2018). We have already cited the study in the paragraph above. In this study, we found that changing any one of the NF-κB binding sites had no significant effect on the transcriptional activity of CMV enhancer/promoter. Maybe this is the reason why the CMV enhancer/promoter always shows high transcriptional activity in various cell lines. Different transcription factor binding sites and transcription factors show synergistic reaction, which jointly contributed to the strong transcriptional activity of CMV enhancer/promoter. Due to its harbouring of many binding sites of multiple transcription factors, it is very difficult to create a mutation of this enhancer/promoter to get a mutant CMV enhancer/promoter that has significant impaired transcriptional activity relative to the wild-type CMV enhancer/promoter.

In this study, we focused on verifying the feasibility and reliability of activating expression of both exogenous and endogenous genes by CRISPR-based trans-enhancer, not focused one verifying a function of a known or potential enhancer. Therefore, we did not used a mutated CMV enhancer/promoter.

When transfecting cells, we included many necessary transfections as controls. All transfections of a cell line were performed simultaneously at the same condition for the sake of comparison.

We are sorry that we have not used any enhancer fragments other than CMV due to the reasons described above. However, your suggestion is very good. We expect to measure other enhancer fragments in future study.

In Figure 2 and Figure 3 as well as Figure 2—figure supplement 2, Figure 2—figure supplement 3, Figure 2—figure supplement 4 the results of the ZsGreen assays are absolutely not clear. What are we supposed to compare, how by looking at the figures can we assess the activation in the different conditions? Also, in the text, there are no precise figures given concerning the fold activation in each condition. In Figure 4A, there are only 2 transfections so there are no robust statistics concerning activation. The reasons for assaying ZsGreen rather measuring RNA output directly are not well justified. All of these figures have to be modified such that the reader can readily assess the differences in each experimental condition. RNA should be measured by RT-qPCR and fold changes and statistics from replicate experiments should be provided.

As suggested by another reviewer, all FACS data were presented as dot plots (FSC vs. ZsGreen) as suggested. Please see the revised Figure 2 and Figure 3 as well as Figure 2—figure supplement 2, Figure 2—figure supplement 3, Figure 2—figure supplement 4. An important sentence was added in the legends of these figures: “The reporter gene activation efficiency was indicated by the percentage of cells with green fluorescence over the threshold (cells in Q1-UR quadrant).” In these experiments, we aimed at evaluating the gene activation capability by using a reporter construct, ZsGreen.

Figure 4A was revised, a further biological replicate was performed and the statistics were added. Please see the revised Figure 4.

Because the expression of reporter gene of ZsGreen can be measured at protein level by detecting fluorescence, we showed gene activation by fluorescence images and fluorescence intensity analyzed by flow cytometry. Therefore, we did not measured expression of reporter gene ZsGreen at mRNA level.

However, in activating ten endogenous genes in seven different cell lines, we detected the gene expression at mRNA level by using qPCR. In all activation of endogenous genes, the fold changes and statistics from three biological replicates were provided. Please the revised Figure 5 and Figure 6.

In Figure 5 and Figure 6 the number of biological replicates and the statistical tests used should both be indicated. The text describing these figures is somewhat oversimplified as they claim that activation of HNF4A results in strong induction of liver function genes. In fact, some are strongly induced, but for others the effect is very low and this is not recognised in the text. Statistics should be shown for the cell cycle analyses in Figure 6C number of replicates and statistical test used.

As to Figure 5 and Figure 6, the experiments were re-performed in the past three months. Three biological replicates were performed for the transfection of each cell line. The statistical difference was analyzed by Student *t* test. The new results were showed as the revised Figure 5 and Figure 6. The figure legends were also revised. Please see the revised manuscript. Please note that Figure 6C is now Figure 7A.

The content of subsection “Statistical analyses” was also revised as follows:

“Each cell transfection for detecting gene expression activation by trans-enhancer was performed in three biological replicates. In each biological replicate, at least three technical replicates were performed. The mean RQ value of technical replicates was used as the RQ value of one biological replicate. The mean RQ value of three biological replicates were used to calculate the final mean and standard deviation (SD). Data were analyzed by Student *t* test when comparing 2 groups. Data were expressed as mean ± SD and differences were considered significant at *P* < 0.05.”

Figure 6C is now Figure 7A. Statistics was shown for the cell cycle analyses in this figure. Number of replicates and statistical test used were added.

Figure S9. There is no obvious difference in the number of cells in the different conditions. This must be quantitated and analysed to provide reliable statistics.

The images of acridine orange-stained cells were counted with ImageJ software. The difference among different treatments were tested with Students t test. There is a significant difference between treated cells and control cells. Please see the added Figure S9C.

In summary, while this paper describes an interesting and novel technique, the data as they are presented require additional replicates, statistical test and clearer presentation to be acceptable.

Yes, three biological replicates were performed for the transfection of each cell line. The statistical difference was analyzed by Student’s t-test. The new results were showed as the revised Figure 4, Figure 5, Figure 6 and Figure 7. The figure legends were also revised. Please see the revised manuscript.

Additional data files and statistical comments:The number of replicates and the statistical tests used are not indicated and more rigorous analyses have to be performed and present in any revised version. The present version is way below acceptable standards.

Each cell transfection for detecting gene expression activation by trans-enhancer was performed in three biological replicates. In each biological replicate, at least three technical replicates were performed. In the detection of gene expression by qPCR, the mean RQ value of technical replicates was used as the RQ value of one biological replicate. The mean RQ value of three biological replicates were used to calculate the final mean and standard deviation (SD). Data were analyzed by Student’s t-test when comparing 2 groups. Data were shown as mean ± SD and differences were considered significant at P < 0.05.

Reviewer #2:

*[…] Although the strategy proposed here is original, the sCMV component needs to be provided in trans (co-transfected), therefore application of this system is limited to cells in culture and would be very challenging to use this strategy* in vivo.

This is a good point. It is clear that the csgRNA-sCMV-based trans enhancer technique is helpful for in vitro application, such as in vitro cell reprogramming and gene activation for gain-of function. However, it’s true that the csgRNA-sCMV-based trans enhancer still faces difficulty in in vivo application. The trans enhancer used a linear CMV enhancer DNA fragment that has a single-stranded overhang complementary to the 3′ end of csgRNA. It is difficult to produce this kind of trans enhancer DNA in the in vivo cells unless transfecting the in vitro pre-prepared trans enhancer with nanoparticle gene carriers together with expression vectors of dCas9-VP64 and csgRNA. However, the current trans enhancer can’t be brought into the in vivo cells by the current most effective in vivo transgenic vector virus (e.g. AAV) that has been approved by FDA to clinical application.

To address the problem, we also developed a new strategy by using a GAL4-UAS system, in which dCas9-VP64-GAL4, sgRNA and UAS-CMV were used. The UAS-CMV was recruited to target gene by dCas9-VP64-GAL4 via the interaction between GAL4 and UAS. We showed that UAS-CMV in both linear and circular forms could be recruited by dCas9-VP64-GAL4 and functioned. This system allows easy in vivo application of the CRISPR-assisted trans enhancer technique. For example, all components including dCas9-VP64-GAL4, sgRNA and LUAS-CMV could be easily provided to in vivo cells as Adeno-associated virus (AAV), a current widely used safe virus vector of human gene therapy.

Thanks for your comments.

A major flaw of this study is that transfection efficiency cannot be determined. It is impossible to know the percentage of cells that are expressing dCas9-VP64 and the proportion of these in which transcriptional activation has occurred. Therefore, the transcriptional effect observed is either under- or over-estimated.

All FACS data were presented as dot plots (FSC vs. ZsGreen) as suggested. Please see the revised figures. All FACS data presented as dot plots of a particular cell line simultaneously transfected by various vectors used the same gating for comparing. Maybe these dot plots tell something about transfection efficiency. It was clear the ZsGreen activation rate is low in our transfection. The main reason for this is the close relationship between the co-transfection and the three independent vectors, a csgRNA expression vector, a dCas9-VP64/VPR expression vector, and a reporter gene ZsGreen expression vector.

However, we think that transfection efficiency do not affect our conclusion, that is, the trans-enhancer itself can activate the reporter gene expression by complexing with dCas9/csgRNA and dCas9-VP64/csgRNA (please see Figure 2 and Figure 3, Figure 2—figure supplement, Figure 2—figure supplement3, Figure 2—figure supplement 4, Figure 3—figure supplement 1, and Figure 3—figure supplement 2). Moreover, trans-enhancer had significant higher gene activation efficiency by complexing with dCas9-VP64/csgRNA than dCas9-VP64/sgRNA (please see Figure 2 and Figure 3, Figure 2—figure supplement, Figure 2—figure supplement3, Figure 2—figure supplement 4, Figure 3—figure supplement 1, and Figure 3—figure supplement 2). Because each cell was simultaneously transfected by various vectors at the same condition and many necessary controls were performed at the same time, the transfection efficiency is not a variable to affect the results of comparison of various transfections. Additionally, in flow cytometry assay, as many as ten thousand cells were measured for each transfection.

Moreover, the reporter gene is only one way to evaluate the gene activation capability. Our major focus is to activate endogenous genes by CRISPR-assistant trans-enhancer. We activated as many as ten endogenous genes in as many as seven different cell lines from various tissues. By comparing with dCas9-VP64/sgRNA in all transfections, it is clear that the trans-enhancer effectively activated these endogenous genes by complexing with dCas9-VP64/csgRNA, with significant high activation efficiency in all transfections than dCas9-VP64/sgRNA. Additionally, dCas9-VP64/sgRNA system included a two vector co-transfection, but the trans-enhancer system included three vectors. The former should have higher transfection efficiency than the latter; however, it still showed significant lower activation efficiency than trans-enhancer.

There are many variables in the experiment, as 4 constructs need to be co-transfected (reporter, dCas9-VP64, csgRNA and sCMV). The experiments should have been done in a cell line stably transfected with the reporter construct and using a single plasmid co-expressing the csgRNA together with mCherry-dCas9-VP64 (or mCherry-dCas9-VPR). This would not only reduce the number of variables but would allow the authors to: (1) Determine the transfection efficiency and the frequency of mCherry+ cells (expressing dCas9-VP64) in which transcriptional activation has occurred (ZsGreen+) and (2) Determine the transactivation level by scoring the MFI of ZsGreen in the mCherry+ ZsGreen+ population.

Yes, this is a good point and if we performed the experiments as suggested, the results should be better. However, as we described above, the transfection efficiency does not affect our conclusion. We focused on determining the gene activation efficiency by comparing various transfection obtained at the same condition. We did not focus on investigating transfection efficiency of different vectors. We think that at the same transfection conditions, the transfection efficiency did not affect our results and conclusion. Also, as we described above, the reporter gene is only one way to evaluate the gene activation capability. Our major focus is to activate endogenous genes by CRISPR-assistant trans-enhancer. We activated as many as ten endogenous genes in as many as seven different cell lines from various tissues. By comparing with dCas9-VP64/sgRNA in all transfections, it is clear that the trans-enhancer effectively activated these endogenous genes by complexing with dCas9-VP64/csgRNA, with significant high activation efficiency in all transfections than dCas9-VP64/sgRNA.

The FACS data should be presented as dot plots (FSC vs.ZsGreen). This allows to visualize the ZsGreen+ population, which is not clear at all (given the low percentage of ZsGreen+ cells) from the histograms.

Yes, all FACS data were presented as dot plots (FSC vs. ZsGreen) as suggested. Please see the revised figures.

The manuscript is poorly written and is full of strange words and expressions and should be proof-read by an English-speaking person.

The whole manuscript was carefully revised to improve the writing.

Figure legends do not describe the figures and should be re-written. Legend for Figure 1B is missing.

The figure legends were carefully revised to describe the figures. Legend for Figure 1B was added. Thank you.

On all FACS data shown, the gating for ZsGreen is not the same throughout (i.e. compare panels in Figure 2).

Yes, all FACS data were presented as dot plots (FSC vs. ZsGreen) as suggested. In all FACS data of a cell, the same gating was used for comparing.

It is not clear how the MFI intensity in Figure 4 was determined. Therefore, it is impossible to assess the validity of the comparisons.

Figure 4 was revised. The MFI intensity was determined by flow cytometry. Please see the revised Figure 4 legends.

In Figure 1B, the percentage of cleavage should be displayed, as it is clear that the csgRNA is not as efficient as the unmodified scaffold. Also, the input is missing.

The legend for Figure 1B and the input band were added, thank you. Yes, it seems that csgRNA is not as efficient as the unmodified scaffold. This is not a quantitative assay. This assay just showed that csgRNA is functional.

In Figure 5, a negative control without the csgRNA is missing. The relative mRNA induction should be calculated relative to this control, as it would directly show the fold induction triggered by dCas9-VP64/csgRNA/sCMV. Also, the y axis should be labelled relative mRNA (and not RNA) expression.In the qPCR data shown in Figure 6A and 6B, it is not indicated what was used to determine the relative expression. Same as in Figure 5, the negative control (without the csgRNA) is missing and should be performed in order to determine the relative expression of each gene. Also, the expression level of the target gene should be shown.

In the transfection of reporter construct, we used a negative control without the csgRNA which revealed that no target gene was activated. Therefore, in the latter transfections to activate endogenous genes (Figure 5), we did not include this control. We focused on comparing target activation efficiency of three different transfection to show the effectiveness of dCas9-VP64/csgRNA/sCMV.

As to Figures 5 and 6, the experiments were re-performed in the past three months. Three biological replicates were performed for the transfection of each cell line. The statistical difference was analyzed by Student’s t-test. The new results were showed as the revised Figure 5 and Figure 6. The figure legends were also revised. Please see the revised manuscript.

The content of subsection “Statistical analyses” was also revised as follows:

Each cell transfection for detecting gene expression activation by trans-enhancer was performed in three biological replicates. In each biological replicate, at least three technical replicates were performed. The mean RQ value of technical replicates was used as the RQ value of one biological replicate. The mean RQ value of three biological replicates were used to calculate the final mean and standard deviation (SD). Data were analyzed by Student *t* test when comparing 2 groups. Data were expressed as mean ± SD and differences were considered significant at *P* < 0.05.

For clarity, a diagram of the reporter construct should be shown.

The whole sequences of HNF4a promoter report vector have been provided and clearly show the reporter construct.

Additional data files and statistical comments:Throughout the manuscript, it is not clear how many times the experiments were done. For example, the qPCR data shown in Figure 5, do the error bars correspond to technical or biological replicates? Is a representative experiment shown? Also, in several figures it is not indicated which samples are compared when the * is shown for statistical significance.

Figures 4, 5, 6, and 7 were revised by performing more biological replicates. In the figure legends, the numbers of biological replicates were given.

Reviewer #3:[…] 1) The authors initially validate their approach (Figure 2) in 293T cells by co-transfecting an exogenous GFP-based reporter construct. Transient transfection efficiencies of 293T cells are known to be extremely high. Why does e.g. dCas9-VP64/csgRNA2-sCMV achieve only 11% of GFP+ cells, assuming that >90% of 293T are transfected?

In our lab, we often transfected 293T cells by a single plasmid expressing EGFP under the control of strong CMV promoter, in which we found that the EGFP activation in cells is about 20~30. In comparison, transfection of dCas9-VP64/csgRNA2-sCMV achieved 11% of GFP+ cells. It is ideal in our opinion. The decreased transient transfection efficiencies of 293T cells in this study have a close relationship with the co-transfection with four independent vectors: a csgRNA expression vector, a dCas9-VP64/VPR expression vector, a reporter gene ZsGreen expression vector, and sCMV. Co-transfection can’t achieve high transfection efficiencies. By combining dCas9-VP64 and csgRNA into one vector, the gene activation efficiency can be further improved in the future.

However, we think that transfection efficiency do not affect our conclusion, that is, the trans-enhancer itself can activate the reporter gene expression by complexing with dCas9/csgRNA and dCas9-VP64/csgRNA (please see the Figure 2 and Figure 3, Figure 2—figure supplement, Figure 2—figure supplement3, Figure 2—figure supplement 4, Figure 3—figure supplement 1, and Figure 3—figure supplement 2). Moreover, trans-enhancer had significant higher gene activation efficiency by complexing with dCas9-VP64/csgRNA than dCas9-VP64/sgRNA (please see the Figure 2 and Figure 3, Figure 2—figure supplement, Figure 2—figure supplement3, Figure 2—figure supplement 4, Figure 3—figure supplement 1, and Figure 3—figure supplement 2). Because each cell was simultaneously transfected by various vectors at the same condition and many necessary controls were performed at the same time, the transfection efficiency is not a variable to affect the results of comparison of various transfections. Additionally, in flow cytometry assay, as many as ten thousand cells were measured for each transfection.

Moreover, the reporter gene is only one way to evaluate the gene activation capability. Our major focus is to activate endogenous genes by CRISPR-assistant trans-enhancer. We activated as many as ten endogenous genes in as many as seven different cell lines from various tissues. By comparing with dCas9-VP64/sgRNA in all transfections, it is clear that the trans-enhancer effectively activated these endogenous genes by complexing with dCas9-VP64/csgRNA, with significant high activation efficiency in all transfections than dCas9-VP64/sgRNA. Additionally, dCas9-VP64/sgRNA system included a two vector co-transfection, but the trans-enhancer system included three vectors. The former should have higher transfection efficiency than the latter; however, it still showed significant lower activation efficiency than trans-enhancer.

2) Regarding Figure 5 and along the line of my previous comment. How do we have to interpret the fold-changes in expression of endogenous genes having seen that in Figure 2 only a comparably small proportion of cells activated the reporter construct? Based on the methods description, all endogenous gene expression analyses were done using the bulk cell cultures. Are the changes in gene expression seen in the bulk culture caused by extremely high transactivation in a small fraction of cells or by homogenous induction across all cells? At least the first scenario is suggested by the pilot experiments shown in Figure 2. Thus, single cell-based approaches could clarify this for the endogenous target genes, e.g. immunofluorescence/FACS for the proteins encoded by the transactivated target genes.

As we described above, the decreased transient transfection efficiencies of 293T cells in this study (Figure 2) have a close relationship with the co-transfection with four independent vectors. Co-transfection can’t achieve very high transfection efficiencies. However, in Figure 5, cells were only transfected by three vectors, a csgRNA expression vector, a dCas9-VP64/VPR expression vector, and sCMV. The transfection was enhanced. Moreover, in Figure 2, we detected reporter gene expression at protein level, but in Figure 5 and Figure 6 we detected gene expression at transcription level by qPCR that has high detection sensitivity due to amplification. The reporter gene expression at protein level was quantitatively detected by flow cytometry, which has relative low detection sensitivity in comparison with qPCR because it has relative low fluorescence detection limitation.

We detected gene expression of target endogenous genes by qPCR. This is a standard approach for detecting gene expression. In many other previous reports of activation gene expression by CRISPR-based methods, the same qPCR detection of gene expression in the bulk culture were widely used. The related references were systematically cited in the Introduction. We found no reported single cell-based approaches were used in this aspect.

Based on the qPCR detection, all endogenous gene expression analyses were done using the bulk cell cultures. “Are the changes in gene expression seen in the bulk culture caused by extremely high transactivation in a small fraction of cells or by homogenous induction across all cells?”. This is a good question. We reduced that this is the changes in gene expression seen in the bulk culture should be caused by the enhanced transactivation in increased fraction of cells. Please see Figure 4B, in which we found that in comparison with other transfections, the transfection of dCas9-VP64/csgRNA and sCMV (DVSC) significantly increased numbers of cells with certain fluorescence intensity threshold. At the highest fluorescence intensity threshold (6.9), no cells were in other transfection, however, there are still about 2% of cells with this highest fluorescence. In flow cytometry, we measured 10000 cells for each transfection, 2% means that as many 200 cells had the highest fluorescence (i.e. target gene ZsGreen expression at protein level). This is very useful. For example, in cell reprogramming, high target gene activation can increase the efficiency of cell reprogramming.

Of course, as suggested by reviewer, it is better to characterize the endogenous target gene activation with single cell-based approaches, e.g. immunofluorescence/FACS for the proteins encoded by the transactivated target genes.

3) If transcriptional activation is achieved only in a fraction (approximately 10%) of the cells despite high transfection efficiencies, then the approach by Xu et al., would have some limitations, at least for bulk cell culture analyses, in comparison to SAM and other methods, which can be also efficiently delivered to target cells by viruses. Where do the authors then see the advantage of their approach, as the requirement for transient transfection remains a drawback, in particular in cell models that are difficult to transfect?

Yes, this study tested the idea that the exogenous and endogenous genes could be efficiently activated by a CRISPR-assistant trans-enhancer in many cells. To our knowledge, this is the first time it’s been shown that genes in cells could be activated by an exogenous enhancer in trans format. We show that genes in cells could be activated by the interaction between genomic DNA and exogenous DNA via protein mediation. This technique has potential application in vitro such as cell reprogramming and cell-based gene therapy.

It is clear that the csgRNA-sCMV-based trans enhancer technique is helpful for in vitro application, such as in vitro cell reprogramming and gene activation for gain-of function. However, it is true to say that the csgRNA-sCMV-based trans enhancer still faces difficulty in in vivo application. The trans enhancer used a linear CMV enhancer DNA fragment that has a single-stranded overhang complementary to the 3′ end of csgRNA. It is difficult to produce this kind of trans enhancer DNA in the in vivo cells unless transfecting the in vitro pre-prepared trans enhancer with nanoparticle gene carriers together with expression vectors of dCas9-VP64 and csgRNA. However, the current trans enhancer can’t be brought into the in vivo cells by the most effective current in vivo transgenic vector virus (e.g. AAV) that has been approved by FDA to clinical application.

To address the problem, we also developed a new strategy by using GAL4-UAS system, in which the dCas9-VP64-GAL4, sgRNA and UAS-CMV were used. The UAS-CMV was recruited to target gene by dCas9-VP64-GAL4 via the interaction between GAL4 and UAS. We should that UAS-CMV in both linear and circular forms could be recruited by dCas9-VP64-GAL4 and functioned. This system allows easy in vivo application of the CRISPR-assisted trans enhancer technique. For example, all components including dCas9-VP64-GAL4, sgRNA and LUAS-CMV could be easily provided to in vivo cells as Adeno-associated virus (AAV), a current widely used safe virus vector of human gene therapy.

Thanks for your comments.

[Editors' note: the author responses to the re-review follow.]

The manuscript has been improved but there are some remaining issues that need to be addressed before acceptance, as outlined below:In this new version, the authors addressed many of the issues raised on the previous versions. They have provided data on the reproducibility of the results with the appropriate statistics. They have used alternative regulatory elements in place of CMV and they have developed an alternative strategy for recruiting the trans-acting element using the GAL4 DNA binding domain. These new innovations have strengthened the study. However, the paper is now too long with too many figures. The paper would be more appropriate if it was shortened to a Tools and Resources article rather than a full manuscript. This can be done by reorganizing the figures. For example, rather than showing all of the fluorescence images and sorting results, this could all be shown in graphical form as is already the case for Figures 8C, D and E and Figure 9C, D and E. This would save a lot of space and condense all the information into a single figure, showing perhaps a single representative sorting experiment and cell images as supplemental data. Also Figure 5 could be shortened to show only 1 or 2 cell types with the rest shown as supplemental data. I also propose that the data on cell cycle changes and migration be removed to shorten the text. This data is not central to the main message of the paper. On the other hand, supplemental figure 11 should be shown in the main figures possibly as an addition to Figure 1.

The paper was thoroughly and carefully revised, significantly shortened (from 9952 words to 6753 words including references). The text of revised manuscript (all contents except references) has 4293 words. After revision, only 6 figures were kept in text due to their central importance to the main message of the paper. The figures were also reorganized.

Finally, the writing is poor and the meaning of what is written is in several places very difficult to understand. Mitigating this will require a major effort.

The paper was thoroughly and carefully revised.